# Cross-Sectional Analysis of Factors Predicting Food Assistance Stigma

**DOI:** 10.3390/bs15070897

**Published:** 2025-07-01

**Authors:** Frances Hardin-Fanning, Ratchneewan Ross, Shuying Sha

**Affiliations:** School of Nursing, University of Louisville, Louisville, KY 40292, USA; ratchneewan.ross@louisville.edu (R.R.); shuying.sha@gmail.com (S.S.)

**Keywords:** food security, food assistance, stigma, self-reliance

## Abstract

The stigma associated with food assistance can be a barrier to resolving food insecurity. Self-reliance expectations likely contribute to this stigma. Aim: This cross-sectional study identified factors that predict perceptions of food assistance stigma. Demographics (age, sex, race, and ethnicity) were collected, and food assistance stigma (Food Resource Acceptability) and self-reliance (Self-Reliance Scale) were measured via REDCap questionnaires from 531 online participants. The research volunteer repository, Research Match, was used for recruitment. Multiple regression was conducted to determine food assistance stigma predictors. Older age, being male, and reporting higher self-reliance significantly predicted the likelihood of stigmatizing food assistance. The social expectation of self-reliance in adulthood contributes to an assigned and anticipated stigma associated with accepting food assistance. This stigma permeates many regions, particularly the United States, and likely contributes to unresolved food insecurity despite the availability of multiple food assistance resources. Future qualitative research should be conducted among older individuals and males with high levels of self-reliance to gain a deeper understanding of how food assistance stigma could be lessened so that appropriate stigma reduction interventions could be tested among this target group.

## 1. Introduction

The stigma associated with food assistance is prevalent in U.S. society, and for those experiencing food insecurity, the fear of being judged by others often becomes a major obstacle in addressing their needs ([9]). Emerging evidence shows that the anticipated stigma associated with needing food assistance contributed to the increase in poor mental health outcomes experienced during the COVID-19 pandemic ([3]). Often grounded in the social expectations of parents to provide for children, a fear of “taking food from those more in need,” and feelings of dependence, individuals experiencing food insecurity report feelings of embarrassment, shame, degradation, humiliation, and being “lower in the social hierarchy”. Food assistance stigma negatively impacts identity, image, reputation, self-worth, and dignity ([26]).

[8] ([8]) developed the Stigma and Food Inequity Conceptual Framework to inform research and policy related to food inequities, including food insecurity and the sociocultural stigma associated with food assistance. Within the framework, the stigma associated with food assistance exists in both individuals who experience food insecurity and in food-secure community members, resulting in those experiencing food insecurity often anticipating stigmatization from others. The manifestations of stigma may be explicit or implicit. At the individual level, they often take the form of prejudice, stereotypes, and discrimination. Stigma also appears at the structural level, evident in policies, marketing practices, and infrastructure ([8]).

At the core of food assistance stigma is stereotype threat (i.e., the perception of confirming a stereotype about one’s group and displaying confirming behaviors based on the stereotype) ([8]). Enacted, anticipated, and internalized stigma progress to this stereotype threat. Enacted stigma is based on previous experiences, actual or perceived, with prejudice and discrimination from others. Anticipated stigma is the expectation, based on previous experiences, of having to endure prejudice and discrimination in the future. Anticipated stigma is typically the rationale for hesitation to accept food assistance. Internalized stigma is the endorsement of negative beliefs about oneself and one’s group, along with the application of these beliefs to oneself, resulting in stereotype threat. The internalization of stigma results in negative behavioral, mental, and physical health outcomes ([8]). Despite public health and food assistance advocacy efforts to ensure confidentiality and non-judgmental interactions, the anticipated stigma related to food assistance typically manifests as the confirming behavior of not accepting food assistance when necessary ([26]).

Globally, food insecurity is associated with greater risks of depression and feelings of stress ([30]), cardiovascular disease and metabolic disorders ([7]; [29]), cholera ([10]), low birth weight ([5]), and cancer mortality ([25]). Given the synergistic impact of food insecurity and food assistance stigma on health outcomes and the critical need to improve food security in multiple populations, it is important to determine the etiology of stigma specific to food assistance. The expectation of self-reliance (i.e., the ability to depend upon oneself and one’s capabilities) ([36]) is highly prevalent in the U.S. and contributes to food assistance stigma ([8]). In populations where high degrees of self-reliance are considered cultural norms, even the assessment of childhood food insecurity in food-secure households can be associated with stigma and perceptions of prejudice ([19]). In addition, the perception of the lower quality of food items provided by assistance organizations being imposed upon rather than chosen by recipients increases food assistance stigma ([9]).

Reducing both the actual and anticipated stigma associated with food assistance is crucial for addressing food insecurity. However, there has been little research related to factors that can be targeted to reduce food assistance stigma ([8]). Since food assistance stigma is often anticipatory and reflective of societal expectations, it is important to understand characteristics and modifiable factors that are associated with this stigma in both food-insecure and food-secure individuals.

### Purpose

The purpose of this study was to identify factors (i.e., age, sex, race, ethnicity, and self-reliance) associated with food assistance stigma. The Strengthening the Reporting of Observational Studies in Epidemiology (STROBE) guidelines were used in the study design and reporting of the results. We hypothesized that participants with higher self-reliance scores would perceive greater food assistance stigma.

## 2. Materials and Methods

### 2.1. Design/Setting

This cross-sectional study was conducted in September and October of 2022 using online recruitment and data collection.

### 2.2. Participants

Participants (N = 531) were recruited through Research Match (RM) ([21]), an online research volunteer repository. Participants had to be at least 18 years of age. The REDCap questionnaire was written in English ([22]).

Individuals self-enroll in the RM repository by creating a password-protected account and answering contact, demographic, and health questions. All 50 U.S. states and Puerto Rico are represented in the RM repository. Two-thirds of RM volunteers are female. Race and ethnicity representation in the RM repository reflects similar U.S. population percentage for Whites (71.3% vs. 75.5%), Asians (4% vs. 6.3%) and American Indian/Alaskan Native (0.9% vs. 1.3%), with an overrepresentation of Native Hawaiians/Pacific Islanders (0.6% vs. 0.3%) and an underrepresentation of Hispanic/Latino (9.3% vs. 19.1%) and Black/African American (10.8% vs. 13.6%) ([31]; [34]).

An email was sent to eligible RM volunteers describing the study’s purpose and participants’ rights. Research Match randomly selected eligible volunteers during the recruitment process. An unsigned, IRB-approved preamble consent form was included in the email invitation, which also included a statement indicating that completion implied consent. The preamble consent also prefaced the REDCap survey. Volunteers consented by clicking on the consent link at the bottom of the RM study webpage, which directed participants to the REDCap questionnaire. There were no tangible participant incentives for completion of the survey. Data were downloaded into Excel spreadsheets with no participant identifiers.

### 2.3. Variables

#### 2.3.1. Demographics

Age, sex, race, and ethnicity were included in these data because of a variety of geographical differences in food assistance participation based on those variables ([13]; [20]; [33]). In addition, intersectional disparities in these demographic characteristics exponentially contribute to the stigma associated with food insecurity, food inequities, and food assistance participation ([8]). In our previous work, we found that being the male head of the household and having children were associated with food assistance stigma in individuals experiencing food insecurity in rural Appalachia ([15]; [18], [19]). Food insecurity disproportionately affects and has greater consequences on children and the elderly ([12]; [4]). Males are 40% more likely than females to report experiencing food assistance stigma when applying for Supplemental Nutrition Assistance Program funds ([23]). The power dynamics associated with social positions of race (e.g., minorities), gender, and class (e.g., older adults) result in additional layers of disadvantage and anticipated stigma ([32]). These disadvantages can also be at the structural or the individual level ([8]).

#### 2.3.2. Food Assistance Stigma

To address the research gap in the measurement of perceived food assistance stigma, we previously developed and evaluated the psychometric properties of the Food Resource Acceptability Questionnaire (FRAQ). Food assistance stigma was defined as negative perceptions and characterizations of people based on characteristics, specifically the stigmatization of participation in means-tested programs ([11]). The FRAQ is a 17-item 4-point Likert-type scale (Cronbach’s alpha 0.89) that measures the likelihood of individuals perceiving food assistance as socially and culturally acceptable ([16]). Potential responses range from 17 to 68. The FRAQ was developed using the behavioral sciences phase guidelines of item development (i.e., identification of the domain and item generation and consideration of content validity); scale development (i.e., pre-testing questions, sampling and questionnaire administration, item reduction, and extraction of latent factors) ([6]); and scale evaluation (i.e., tests of dimensionality, reliability, and validity) ([2]). The FRAQ consists of two subscales, including (1) food assistance stigma perception (Cronbach’s alpha 0.84), which explained 25.4% of the item variance, and (2) food as a basic right (Cronbach’s alpha 0.76), explaining an additional 21.4% of the variance. Items on the “food as a basic right” subscale are reverse-scored, and the total FRAQ score reflects the perception of food assistance as socially acceptable and without stigma. Lower FRAQ scores reflect greater food assistance stigma ([16]). Cronbach’s alpha for the FRAQ in this sample was 0.88.

#### 2.3.3. Self-Reliance

The Self-Reliance Scale is a 3-item short scale adapted by [27] ([27]) from the 25-item full-length Resilience Scale ([36]). Items are scored on a 7-point Likert scale ranging from disagree (1) to agree (7), with possible scores ranging from 3 to 21. Higher scores reflect a higher degree of self-reliance. The three items were loaded on a unique factor in a large-battery study (loading range from 0.73 to 0.77) and demonstrated moderate precision, and alpha = 0.80 ([27]). Cronbach’s alpha for the Self-Reliance Scale in this sample was 0.77.

### 2.4. Data Sources/Measurement

The data were collected using a REDCap ([22]) questionnaire (Appendix A) ([16]), which was accessible via emails from the Research Match repository. Following receipt of a recruitment email from Research Match, participants clicked through the University of Louisville Biomedical Institutional Review Board and approved (22.0725) consent to access the questionnaire. Data were downloaded into Excel files and included no participant identifiers.

### 2.5. Bias

The sole inclusion criterion of age potentially reduced bias. Response and acquiescence bias were addressed by having both negatively and positively worded items on the FRAQ.

### 2.6. Analysis

Descriptive statistics, including frequency distribution and central tendency measures (mean, SD, and range), were used to describe participant characteristics and key study variables. Bivariate statistics (i.e., Pearson correlation, ANOVA, independent t-test) were first conducted to examine the relationship between food acceptance. Multiple regression analysis was then conducted to identify variables that predict food assistance when other variables were controlled. All analyses were conducted in SPSS V.29, and statistical significance levels were set at *p* < 0.05 for all analyses.

## 3. Results

### Participant Characteristics

Participants’ demographic information and descriptive statistics of self-reliance and the FRAQ are presented in Table 1. Most participants were White, non-Latino/Hispanic, and female. Age was negatively correlated with FRAQ scores (r = −0.19, *p* < 0.001) but positively correlated with self-reliance (r = 0.18, *p* < 0.001). FRAQ scores were not significantly different between Latino and non-Latino or among Black, White, and Asian. Females were more likely to view food assistance as acceptable (M = 63.41, M = 53.73) (t = 5.26, *p* < 0.001) than males. We first conducted bivariate analyses to explore the relationship between the FRAQ and demographic factors, as well as self-reliance, and the results are presented in Table 2. When dividing the participants into four age groups, the younger group demonstrated significantly higher willingness to accept food assistance (*p* < 0.001). Females (M = 63.41) were more likely to view food assistance as acceptable than males (M = 53.73) (t = 5.26, *p* < 0.001). However, we did not find any significant relationship between race, ethnicity, and FRAQ scores. The Pearson correlation between self-reliance and the FRAQ is significant (r = −0.11, *p* < 0.05), as those who were more self-reliant were less likely to view food assistance as acceptable and without stigma.

The variables (age, sex, self-reliance) showed significant association with the FRAQ and were then entered in a multiple regression model; sex was dummy-coded with male as the reference group. Altogether, the model explained about 10% of the variance in the FRAQ (R2 = 0.096, *p* < 0.001). The regression parameter is presented in Table 3. When other variables were controlled, age was still significantly associated with food assistance stigma (B = −0.10, *p* < 0.001), and females demonstrated a higher average level of food acceptance (B = 5.95, *p* < 0.001). Controlling for demographic variables, self-reliance was still negatively related to food acceptance (B = −0.35, *p* = 0.017).

## 4. Discussion

In this study, we hypothesized that those reporting greater self-reliance would also report higher levels of food assistance stigma. Older age, being male, and reporting higher self-reliance significantly predicted the likelihood of stigmatizing food assistance. Self-reliance expectations are likely contributing to food assistance stigma. Older participants were likely to have a greater sense of food assistance stigma than their younger counterparts in our study. This finding could be explained through theories of successful aging, which posit that greater self-reliance lessens health-seeking behaviors with an unwillingness to seek medical assistance and subsequently results in poorer physical and mental health outcomes ([14]; [24]; [35]). Hence, it is likely that the developmental stage-related traits of desired independence and high self-reliance in late adulthood influenced food assistance stigma in this cohort. 

Two of the items on the Self-Reliance Scale (“When I am in a difficult situation, I can usually find my way out of it,” and “My belief in myself gets me through hard times”.) comprise an expectation that individuals are solely responsible for the resolution of hardships. Given that adult responsibility for basic needs, particularly dietary needs, is often a social expectation, lower self-reliance is congruent with higher levels of stigma associated with food assistance. This is the first study to show the relationship between self-reliance and the stigma of food assistance.

A reduction in food assistance stigma factors sensitive to age, gender, and self-reliance will require interventions at the individual and structural levels ([8]). According to [8] ([8]), interventions at the individual level include education to address stereotypes (e.g., affirmation interventions to reduce stereotype threat and restore self-integrity, counseling to promote healthy food purchasing decisions), intergroup contact to humanize members of stigmatized populations and reduce implicit and explicit prejudice (e.g., social support groups and belongingness interventions to reduce social exclusion), and cognitive dissonance interventions (e.g., expressive writing to strengthen coping with enacted and/or anticipated stigma) that lead perceivers to confront inconsistencies between their beliefs and values. Recommended structural level interventions include mass media campaigns aimed at changing public attitudes toward members of stigmatized groups and cultural competence training to communicate diversity values ([8]).

In our previous qualitative research with college students in Appalachia, participants provided suggestions to reduce food assistance stigma by partnering with other community agencies (i.e., libraries, cooperative extension offices, health departments) and incorporating food assistance into other activities (e.g., recreational events, student activities, reading programs, cooking classes) ([15]). While the goal of these interventions is the normalization of food assistance, stigma has been a longstanding issue in food insecurity resolution and will require a multi-level interventional approach. In two of our earlier studies, the lack of means-testing for food assistance was a significant predictor of whether individuals were willing to accept assistance ([1]; [17]). By providing food items to individuals irrespective of income, there was a greater likelihood of reducing stigma and, subsequently, normalizing food assistance.

### 4.1. Limitations

Self-selection is a potential bias in this study, as the Research Match pool of more than 145,000 research participant volunteers is likely more motivated to complete online studies. However, the sole inclusion criterion for this cross-sectional study was ≥18 years of age, and participation was anonymous to encourage a broad demographic range of participants. Most participants were White females, and additional research is needed to explore the relationship between self-reliance and food assistance stigma in other populations, particularly those in which intersectionality potentiates the health impact of anticipatory stigma. Although more than one-fifth of our sample was male, future studies are needed to specifically determine the impact of self-reliance expectations on male heads of household.

### 4.2. Generalizability

Despite decades of food advocacy and nutrition supplemental programs, food insecurity continues to impact millions of people. The stigma associated with food insecurity and acceptance of food assistance is common in many populations and results in poor health outcomes. The nearly universal expectation of self-reliance in adulthood contributes to this stigma.

## 5. Conclusions

The results from this study can inform future interventions to increase food assistance acceptability and subsequently decrease food insecurity. To better understand how to reduce food assistance stigma, future qualitative research should focus on the development of food assistance stigma reduction interventions among older adults, males, and those with high levels of self-reliance. Self-reliance likely influences the stigma associated with acceptance of food assistance. Also, to facilitate food insecurity resolution, more research, in the context of the Stigma and Food Inequity Conceptual Framework ([8]), is needed to determine additional factors associated with the perception of food assistance as being socially and culturally acceptable. Interventions aimed at reducing individual stigma are also needed to enhance food assistance acceptability and, subsequently, reduce food insecurity.

At the individual level, interventions that promote perceptions that there is no conflict between self-reliance and acceptance of food assistance are needed. The recent pandemic elucidated how rapid changes in socioeconomic circumstances (e.g., natural disaster, life crisis, disability) can create food assistance needs irrespective of prior self-reliance or independence. Behavioral interventions that reframe food assistance as a means of preventing food waste and reducing the resultant climate impact of carbon emissions from food in landfills also have the potential to lessen food assistance stigma.

Changing individual perceptions of stigma associated with food assistance acceptance will be difficult until structural stigma manifestations are removed. Additional research should also be conducted to understand how structural stigma contributes to or potentiates individual stigma. Research partnerships with community agencies (e.g., health departments, cooperative extension offices, local libraries) would facilitate trusting relationships with participants and allow for a more in-depth exploration of other factors associated with food assistance stigma. Structural stigma manifestations at the policy, marketing, and infrastructure levels tend to compound the marginalization of food-insecure residents in lower-income neighborhoods ([8]). The subsequently imposed disparities in food access result in poorer health outcomes for these at-risk communities ([28]). Self-reliance becomes a near-unattainable goal in neighborhoods where economic hardship and deprivation have existed for generations.

## Figures and Tables

**Table 1 behavsci-15-00897-t001:** Participant characteristics.

Variables		N	%
Sex	Male	114	21.5
	Female	417	78.5
Race	Latino	32	6.1
	Non-Latino	496	93.9
Ethnicity	Asian	21	3.9
	White	462	86.7
	Others	15	2.8
		Mean (SD)	Range
Age (years)		51.06 (17.31)	18–92
Self-reliance		17.04 (3.06)	3–21
FRAQ		62.25 (10.33)	30–85

**Table 2 behavsci-15-00897-t002:** Descriptive statistics of the FRAQ by demographic factors; N = 513.

Variable	N	Mean	SD	Min	Max	*p* Value
**Age**						
18–29 yrs	67	65.19	10.09	42	85	<0.001 ^AN^
30–49 yrs	177	63.89	10.09	33	84	
50–69 yrs	203	60.68	10.14	35	85	
70 and older	80	60.11	10.53	30	83	
**Race**						
Latino	32	61.84	9.88	47	84	0.80 ^t^
Non-Latino	491	62.32	10.37	30	85	
**Ethnicity**						
Asian	21	61.81	10.01	46	80	0.61 ^AN^
Black	35	61.57	12.23	33	82	
White	457	62.21	10.21	30	85	
Others	14	65.79	10.05	50	80	
**Sex**						
Male	114	57.73	10.90	33	85	<0.001 ^t^
Female	417	63.41	9.84	30	85	
Total	531	62.24	10.33	30	85	

^AN^: Analysis of Variance (ANOVA); ^t^: independent *t*-test.

**Table 3 behavsci-15-00897-t003:** Multiple regression predicting lowering food assistance acceptability.

Predictors	B	SE	β	*t*	*p* Value
Age	−0.10	0.03	−0.17	−3.99	<0.001
Female	5.95	1.05	0.24	5.65	<0.001
Self-reliance	−0.35	0.15	−0.10	−2.40	0.017

R^2^ = 0.096, F(2,523) = 18.34, *p* < 0.001.

## Data Availability

Data in Excel spreadsheets are available by contacting the corresponding author via email.

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
