# Peer review of "Cross-Sectional Analysis of Factors Predicting Food Assistance Stigma"

_behavsci, 2025, doi:10.3390/bs15070897_

Round 1

Reviewer 1 Report

Comments and Suggestions for Authors

Dear authors,

I am delighted to have had the opportunity to review your manuscript on the fascinating topic of “food assistance stigma.” In my opinion, this is a socially important issue that warrants scientific examination. 

I found the paper's argumentation easy to follow, but I have two somewhat major points of criticism that I believe require revision:

1) The theoretical background to the topic could be a little more in-depth and less “superficial.” As a reader, I would be interested to know, for example, what specific definition of stigma the FRAQ is based on, in which contexts “food assistance stigma” has already been studied, and how the relevance of looking at sociodemographic characteristics in detail is derived.

2) Closely related to point 1, the discussion and conclusions are also rather “superficial.” Here, I would like to see a more in-depth discussion of the possible causes of gender-specific, age-related, and “self-reliance”-related differences. What specific ideas (you have mentioned a few, but beyond that) do you have on how to reduce food assistance stigma, what further implications do you see, and how could these be empirically tested?

Author Response

Thank you for the recognition of the importance of this issue and for the opportunity to clarify/edit the manuscript.  We appreciate the careful attention and time spent reviewing our work.  We have addressed each concern. 

 Reviewer 1

The theoretical background to the topic could be a little more in-depth and less “superficial.” As a reader, I would be interested to know, for example, what specific definition of stigma the FRAQ is based on, in which contexts “food assistance stigma” has already been studied, and how the relevance of looking at sociodemographic characteristics in detail is derived.

INTRODUCTION: We have added a more in-depth explanation of how food assistance stigma develops in individuals who experience food insecurity. We also described the synergies between food assistance stigma and food insecurity in their impact on health outcomes.

MATERIALS AND METHODS: We included our previous work as well as the work of other food insecurity researchers to explain the inclusion of our independent variables.

VARIABLES: We have included the stigma definition used in the development and testing of the FRAQ.

Closely related to point 1, the discussion and conclusions are also rather “superficial.” Here, I would like to see a more in-depth discussion of the possible causes of gender-specific, age-related, and “self-reliance”-related differences. What specific ideas (you have mentioned a few, but beyond that) do you have on how to reduce food assistance stigma, what further implications do you see, and how could these be empirically tested?

DISCUSSION: We have incorporated the answers in the Discussion section of the paper by including both our prior research and the research of others.

CONCLUSION: We have included a sentence specific to community partnerships in exploration of other factors associated with food assistance stigma.

Reviewer 2 Report

Comments and Suggestions for Authors

The manuscript is interesting and brings significant scientific value. The title of the article stands out for its originality, and the authors used an innovative research approach. Particularly noteworthy is the introduction of the self-reliance variable as a potential factor influencing the stigma of food aid. I believe that this publication could have a significant impact on the development of health and social policy, both in the United States and in other countries. The literature review is adequate, although it could have been more extensive. The methodology used and the statistical analysis are clearly presented, and the interpretation of the results and their discussion are appropriate. 

However, the work is not without some shortcomings. According to me, there is a lack of information on the process of recruiting participants - it is not known whether the sample was randomly or non-randomly selected, which raises doubts about the generalizability of the results to the entire population studied. In addition, although the hypothesis focuses on men, they were in fact a minority in the sample, which may affect the accuracy of the conclusions drawn. It would therefore be reasonable to better justify the approach taken.

I also miss practical recommendations - it would have been useful to point out possible actions that could help those hesitant to seek help. It seems to me that the conclusion would also benefit from a more explicit reference to the hypothesis. It would also be good to suggest directions for future research, for example, taking into account other variables or trying to determine which social groups are afraid to use food aid because of stigma (create a profile of such a person). Such a study could in the future for earlier assistance to such people before they are affected by food insecurity and fear of stigmatization.

Author Response

Thank you for the recognition of the importance of this issue and for the opportunity to clarify/edit the manuscript.  We appreciate the careful attention and time spent reviewing our work.  We have addressed each concern. 

Reviewer 2

The literature review is adequate, although it could have been more extensive.

INTRODUCTION: We have expanded the literature review to better describe the development of food assistance stigma as well as the synergistic impact of stigma and food insecurity on health outcomes.

According to me, there is a lack of information on the process of recruiting participants - it is not known whether the sample was randomly or non-randomly selected, which raises doubts about the generalizability of the results to the entire population studied.

METHODS AND MATERIALS: We have explained the recruitment, consenting, and enrollment process in this section.  

In addition, although the hypothesis focuses on men, they were in fact a minority in the sample, which may affect the accuracy of the conclusions drawn. It would therefore be reasonable to better justify the approach taken.

Limitations:  Our hypothesis does not specifically focus on men. However, we have provided the rationale for including biological sex as an independent variable.  We have also included the male percentage of our sample as a limitation and stated the need for future studies specifically targeting male heads of households.

It seems to me that the conclusion would also benefit from a more explicit reference to the hypothesis.

DISCUSSION: We have opened the discussion section with a reference to our hypothesis

I also miss practical recommendations - it would have been useful to point out possible actions that could help those hesitant to seek help. It would also be good to suggest directions for future research, for example, taking into account other variables or trying to determine which social groups are afraid to use food aid because of stigma (create a profile of such a person). Such a study could in the future for earlier assistance to such people before they are affected by food insecurity and fear of stigmatization.

DISCUSSION: We have incorporated our prior research with individuals in Appalachia as well as our study conducted in a large metropolitan hospital. We have also provided relevance to the framework when developing and testing interventions.

Round 2

Reviewer 1 Report

Comments and Suggestions for Authors

Dear authors, thank you very much for revising the manuscript so carefully. I have no further comments :-)